# HOI-Diff: Text-Driven Synthesis of 3D Human-Object Interactions using Diffusion Models

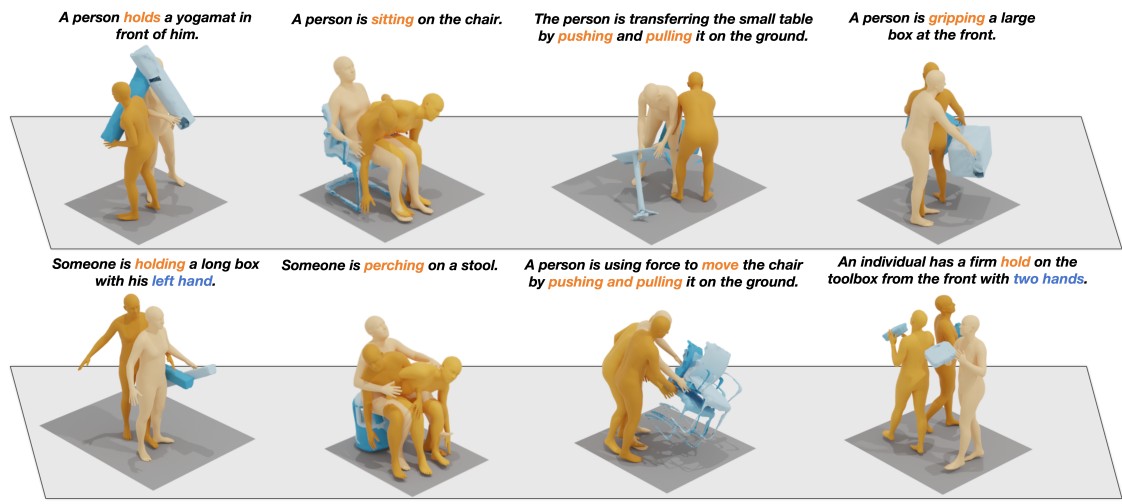

*A person holds a yogamat in front of him.*

*A person is sitting on the chair.*

*The person is transferring the small table by pushing and pulling it on the ground.*

*A person is gripping a large box at the front.*

*Someone is holding a long box with his left hand.*

*Someone is perching on a stool.*

*A person is using force to move the chair by pushing and pulling it on the ground.*

*An individual has a firm hold on the toolbox from the front with two hands.*

Figure 1. **HOI-Diff can generate realistic motions for 3D human-object interactions given a text prompt and object geometry.** Please see the supplementary material for video results. *Darker color indicates later frames in the sequence. Best viewed in color.*

## Abstract

We address the problem of generating realistic 3D human-object interactions (HOIs) driven by textual prompts. To this end, we take a modular design and decompose the complex task into simpler sub-tasks. We first develop a dual-branch diffusion model (DBDM) to generate both human and object motions conditioned on the input text, and encourage coherent motions by a cross-attention communication module between the human and object motion generation branches. We also develop an affordance prediction diffusion model (APDM) to predict the contacting area between the human and object during the interactions driven by the textual prompt. The APDM is independent of the results by the DBDM and thus can correct potential errors by the latter. Moreover, it stochastically generates the contacting points to diversify the generated motions. Finally, we incorporate the estimated contacting points into the classifier-guidance to achieve accurate and close contact between humans and objects. To train and evaluate our approach, we annotate the BEHAVE dataset with text descriptions. Experimental results on BEHAVE and OMOMO demonstrate that our approach produces realistic HOIs with various interactions and different types of objects. Our code and data annotations will be publicly available.

## 1. Introduction

Text-driven synthesis of 3D human-object interactions (HOIs) aims to generate motions for both the human and object that form coherent and semantically meaningful interactions. It enables virtual humans to naturally interact with objects, which has a wide range of applications in AR/VR, video games, and filmmaking, etc.

The generation of natural and physically plausible 3D HOIs involves humans interacting with *dynamic* objects in *various* ways according to the text prompts, thereby posing several challenges. First, the variability of object shapes makes it particularly challenging to generate semantically meaningful contact between the human and object to avoid floating objects. Second, the generated HOIs should be

faithful to the input text prompts as there are many plausible interactions between human and the same object (*e.g*, a person carries a chair, sits on a chair, pushes or pulls a chair). Text-driven 3D HOI synthesis with a diverse set of interactions is not yet fully addressed. Third, the development and evaluation of 3D HOI synthesis models requires a high-quality human motion dataset with various HOIs and textual descriptions, but existing datasets lack either diverse HOIs [13, 26, 37] or detailed textual descriptions with interacting body parts and action [4, 11]. It is important to note that CG-HOI [11] has not made their code or annotations publicly available. In contrast, we will release both our code and annotations.

Current methods cannot fully handle all the challenges. On one hand, recent methods [14, 19, 25, 36, 47, 49, 59, 67] can synthesize realistic human motions for HOIs for *static* objects only. They usually synthesize the motion in the last mile of interaction, *i.e*, the motion between the given starting human pose and the final interaction pose, and overlook the movement of the objects when the human is interacting with them. On the other hand, existing methods for motion generation with dynamic objects do not adequately reflect real-world complexity. For instance, they focus on grasping small objects [12], provide the object motion as conditioning [27], predict deterministic interactions between the human and the same object without the diversity [40, 61], consider only a small set of interactions (*e.g*., sit/lift [25], sit/lie down [14], sit [19, 36, 67], grasp [49, 59]), or investigate a single type of object (*e.g*., chair [19, 67]).

In this paper, we introduce **HOI-Diff** for 3D HOIs synthesis involving humans interacting with different types of objects in diverse ways, which are both physically plausible and semantically faithful to the textual prompt, as shown in Figure 1. Our key insight is to decompose 3D HOIs synthesis into three modules to reduce the complexity of this challenging task. (a) **coarse 3D HOIs generation** that extends the human motion diffusion model [51] to a dual-branch diffusion model (DBDM) to generate both human and object motions conditioning on the input text prompt. To encourage coherent motions, we develop a cross-attention communication module, exchanging information between the human and object motion generation models; (b) **affordance prediction diffusion model** (APDM) that estimates the contacting points between the human and object during the interactions driven by the textual prompt. Our APDM does not rely on the results of the DBDM and thus can recover from its potential errors. Moreover, it stochastically generates the contacting points to diversity the generated motions; and (c) **affordance-guided interaction correction** that incorporates the estimated contacting information and employs the classifier-guidance to achieve accurate and close contact between humans and objects, significantly alleviating the cases of floating objects. Compared with designing a monolithic model, HOI-Diff disentangles motion generation for humans and objects and estimation of their contacting points, which are later integrated to form coherent and diverse HOIs, reducing the complexity and burden for each of the three modules.

For both training and evaluation purposes, we annotate each video sequence in BEHAVE dataset [4] with text descriptions, which mitigates the issue of severe data scarcity for text-driven 3D HOIs generation. In addition, we evaluate our approach on the OMOMO dataset [27], which focuses on the manipulation of two hands. Extensive experiments validate the effectiveness and design choices of our approach, particularly for dynamic objects, thereby enabling a set of new applications in human motion generation.

## 2. Related Work

**Human Motion Generation with Diffusion Models.** The denoising diffusion models have been widely used 2D image generations [39, 43, 44] and achieved impressive results. Recent work [1, 3, 5–7, 20, 42, 45, 48, 51, 52, 58, 60, 64–66, 68] apply the diffusion model in the task of human motion generation. While these methods have successfully generated human motion, they usually generate isolated motions in the free space without considering the objects the human is interacting with. Our method is primarily focused on motion generation with human-object interactions.

**Scene- and Object-Aware Human Motion Generation.** Recent works condition motion synthesis on scene geometry [17, 55, 57, 69]. This facilitates the understanding of human-scene interactions. However, the motion fidelity is compromised due to the lack of paired full scene-motion data. Other approaches p[14, 19, 25, 36, 47, 67] instead focus on the interactions with the objects and can produce realistic motions. However, they focus on interacting with static objects with limited interactions. OMOMO [27] can generate full-body motion from the object motion. The object motion is needed as input in OMOMO, whereas our method can jointly synthesize human motion and object motion. IMoS [12] synthesizes the full-body human along with the 3D object motions from textual inputs, but it only focuses on grasping small objects with hands. InterDiff [61] predicts whole-body interactions with dynamic objects. Note that the interaction type is deterministic. Different from this, we tackle the motion synthesis task, where the interaction with the same object can be controlled by the text prompt. Recently, there has been a surge of interest in the text-driven synthesis of 3D human-object interactions for dynamic objects, resulting in the development of concurrent works [11, 26, 46, 56, 62]. CG-HOI [11] and HOIAnimator [46] uses SMPL parameters as the motion representation, which may result in unsmooth motion due to the potential difficulty in optimization. Instead, we use common skeletal joints similar to most text-to-motion methods, harnessing

the power of pre-trained human motion generation models. Chois Li et al. [26] relies on the initial state and object waypoints to generate HOIs, which reduces motion diversity for both the human and the object. InterFusion [8] and F-HOI [63] generate static 3D HOIs from text description, lacking both human and object motions.

**Affordance Estimation.** The affordance estimation on 3D point cloud is studied in Deng et al. [9], Iriondo et al. [18], Kim and Sukhatme [22, 23], Kokic et al. [24], Mo et al. [31], Ngyen et al. [32]. Overall affordance learning is a very challenging task. Instead of predicting the point-wise contact labels, we simplify it by directly regressing the contact points for human-object interactions, making it more tractable without significantly compromising accuracy.

## 3. Method

The overview of our proposed approach are illustrated in Figure 2. We introduce a dual-branch Human-Object Interaction Diffusion Model (DBDM), which can produce diverse yet consistent motions, capturing the intricate interplay and mutual interactions between humans and objects (Sec. 3.2). To ensure physically plausible contact between humans and objects, we propose a novel affordance prediction diffusion model (APDM) (Sec. 3.3), whose output will be used as classifier guidance (Sec. 3.4) to correct the interactions at each diffusion step of human/object motion generation.

### 3.1. Background

**Motion Representations.** We denote a 3D HOI sequence as $\boldsymbol{x} = \{\boldsymbol{x}^h, \boldsymbol{x}^o\}$. It consists of human motion sequence $\boldsymbol{x}^h \in \mathbb{R}^{L \times D^h}$ and object motion sequence $\boldsymbol{x}^o \in \mathbb{R}^{L \times D^o}$, where $L$ denotes the length of the sequence. For $\boldsymbol{x}^h$, we adopt the redundant representation widely used in human motion generation [13] with $D^h = 263$, which include pelvis velocity, local joint positions, velocities and rotations of other joints in the pelvis space, and binary foot-ground contact labels. For the object motion sequence $\boldsymbol{x}^o$, we assume the object geometry is given as an input, and thus we only need to estimate its 6DoF poses in the generation, *i.e*, $D^o = 6$. We represent each object instance as a point cloud of 512 points $\boldsymbol{p} \in \mathbb{R}^{512 \times 3}$.

**Diffusion Model for 3D HOI Generation.** Given a prompt $\boldsymbol{c} = (\boldsymbol{d}, \boldsymbol{p})$, consisting of a textual description $\boldsymbol{d}$ and the object instance's point cloud $\boldsymbol{p}$, a diffusion model $p_\theta(\boldsymbol{x}_{t-1}|\boldsymbol{x}_t, \boldsymbol{c})$[1] learns the reverse diffusion process to generate clean data from a Gaussian noise $\boldsymbol{x}_T$ with $T$ consecutive denoising steps

$$p_\theta(\boldsymbol{x}_{t-1}|\boldsymbol{x}_t, \boldsymbol{c}) := \mathcal{N}(\boldsymbol{x}_{t-1}, \mu_\theta(\boldsymbol{x}_t, t, \boldsymbol{c}), (1-\alpha_t)\mathbf{I}), \quad (1)$$

where $t$ is the denoising step. Following [51], our diffusion

---

[1]We use superscripts $h$ and $o$ to denote human and object sequence, respectively. Without a superscript, it means the 3D HOI sequence, containing both $\boldsymbol{x}^h$ and $\boldsymbol{x}^o$. Subscript is used for the diffusion denoising step.

model $M_\theta$ with parameters $\theta$ predicts the final clean motion $\boldsymbol{x}_0 = M_\theta(\boldsymbol{x}_t, t, \boldsymbol{c})$.

We sample $\mathbf{x}_{t-1} \sim \mathcal{N}(\boldsymbol{\mu}_t, \Sigma_t)$ and compute the mean as in [33]

$$\boldsymbol{\mu}_t = \frac{\sqrt{\alpha_{t-1}}\beta_t}{1-\alpha_t}\boldsymbol{x}_0 + \frac{\sqrt{1-\beta_t}(1-\alpha_{t-1})}{1-\alpha_t}\boldsymbol{x}_t, \quad (2)$$

where $\alpha_t = \prod_{s=1}^t (1-\beta_s)$ and $\beta_t \in (0, 1)$ are the variance schedule. $\Sigma_t = \frac{1-\alpha_{t-1}}{1-\alpha_t}\beta_t$ [16] is a variance scheduler of choice. Similar to $\boldsymbol{x}_t$, $\boldsymbol{\mu}_t$ consists of $\boldsymbol{\mu}_t^h$ and $\boldsymbol{\mu}_t^o$, corresponding to human and object motion, respectively.

Simply adopting the diffusion model described in Eq.(1) would impose a huge burden on the model, which requires joint generation of human and object motion and more critically, enforcement of their intricate interactions to follow the input textual description. In this paper, we propose **HOI-Diff** for 3D HOIs generation, disentangling motion generation for humans and objects and estimation of their contacting points. They are later integrated to form coherent and diverse HOIs, which reduces the complexity and burden for each of the three modules, leading to better generation performance as evidenced by our experiments.

### 3.2. Coarse 3D HOIs Generation

First, we introduce a dual-branch diffusion model (DBDM) to generate human and object motions that are roughly coherent. As shown in Figure 3, it consists of two Transformer models [54], human motion diffusion model (MDM) $M^h$ and object MDM $M^o$, which work similar to [51]. Specifically, at the diffusion step $t$, they take the text description and noisy motions $\boldsymbol{x}_t^h$ and $\boldsymbol{x}_t^o$ as input and predict clean human and object motions $\boldsymbol{x}_0^h$ and $\boldsymbol{x}_0^o$, respectively.

To enhance the learning of interactions of the human and object when generating their motion, we introduce a Communication Module ($CM$) designed for exchanging feature representations between the human MDM $M^h$ and the object MDM $M^o$. $CM$ is a Transformer block that receives the intermediate feature $\boldsymbol{f}^h, \boldsymbol{f}^o$ from both $M^h$ and $M^o$. It then processes these inputs to generate refined updates based on the cross attention mechanism [54]. The updated feature representations $\tilde{\boldsymbol{f}}_h$ and $\tilde{\boldsymbol{f}}_o$ of the human and object are then conditioned on each other, which are then fed into the subsequent layers of their respective branches to estimate clean human and object motion $\boldsymbol{x}_0^h$ and $\boldsymbol{x}_0^o$, respectively. The $CM$ is inserted at the 4th transformer layer for human MDM and the last layer for object MDM, which was empirically found to work better.

Given the limited data availability for 3D HOI generation, during training, the human motion model $M^h$ finetunes a pretrained human MDM [51]. This fine-tuning is critical to ensure the smoothness of the generated human motions. We ablate this design choice in Sec. 4.3. Object MDM is trained from scratch. We modify the input and output linear layers to take in the object motion which has a different

Figure 2. **Overview of HOI-Diff for 3D HOIs generation using diffusion models.** Our key insight is to decompose the generation task into three modules: (a) coarse 3D HOI generation using a dual-branch diffusion model (DBDM), (b) affordance prediction diffusion model (APDM) to estimate the contacting points of humans and objects, and (c) affordance-guided interaction correction, which incorporates the estimated contacting information and employs the classifier-guidance to achieve accurate and close contact between humans and objects to form coherent HOIs.

dimension from the human motion. More details of DBDM are in Appendix A.1.

### 3.3. Affordance Estimation

Due to the complexity of the interactions between a human and object, DBDM alone usually fails to produce physically plausible results, leading to floating objects or penetrations. To improve the generation of intricate interactions, the problem that needs to be solved is to *identify where the contacting areas are* between the human and object. InterDiff [61] defines the contacting area based on the distance measurement between the surface of human and object. This approach, however, heavily relies on the quality of the generated human and object motions and cannot recover from errors in the coarse 3D HOI results. In addition, the contact area is diverse even with the same object and interaction type, *e.g*, "sit" can happen on either side of a table. To this end, we introduce an Affordance Prediction Diffusion Model (APDM) for affordance estimation. As illustrated in Figure 4, the input includes a text description $d$ and the object point cloud $p$. Our APDM doesn't rely on the results of the DBDM and thus can recover from the potential errors in DBDM. In addition, it stochastically generates the contacting points to ensure the diversity of the generated motions.

Affordance estimation in 3D point clouds itself is a notably challenging problem [9, 18, 22–24, 31, 32], especially in the context of 3D HOI generation involving textual prompt. In this paper, we consider eight primary body joints – the `pelvis`, `neck`, `feet`, `shoulders`, and `hands` – as the interacting parts in HOI scenarios. It can effectively model common interactions such as grasping an object with both hands, sitting actions involving the pelvis and back, or lifting with a single hand. We use binary contact labels to determine which joints are in contact with the object. Subsequently, we predict eight corresponding contact points on the object surface, identified as the points closest to the selected body joints. Note that the binary contact label estimation for different body joints are independent, allowing us to handle complex HOIs.

Specifically, at each diffusion time step $n$ of APDM[2], the noisy data consists of human contact labels representing the contact status for the eight primary body joints, denoted as $y_n^h \in \{0,1\}^8$, and the eight corresponding contact points on the object surface, denoted as $y_n^o \in \mathbb{R}^{8 \times 3}$. The model is designed to predict both contact probabilities and contact positions. Subsequently, dynamic selection of contacting body joints is performed by considering predicted probabilities over a specific threshold $\tau$ (set to be 0.6). The corresponding contact points on the object are then determined based on the selected joints. APDM works similar to the diffusion denoising process described in Eq.(1). Besides, we utilize a large language model (ChatGPT) to determine whether the object state $y_0^s \in \{0,1\}$ should be set to static ($y_0^s = 1$) based on the textual description, which can help us better process static objects when synthesizing 3D HOIs, as discussed in the following section. All the clean affordance data is grouped as $y_0 = (y_0^h, y_0^o, y_0^s)$. More implementation details are in Appendix A.2.

### 3.4. Affordance-guided Interaction Correction

With the estimated affordance, we can better align human and object motions to form coherent interactions. To this end, we propose to use the classifier guidance [10] to achieve accurate and close contact between humans and objects, significantly alleviating the cases of floating objects.

Specifically, in a nutshell, we define an analytic function $G(\mu_t^h, \mu_t^o, y_0)$ that assesses how closely the generated human joints and object's 6DoF pose align with a desired objective. In our case, it enforces the contact positions of human and object to be close to each other and their motions are smooth temporally. Based on the gradient of $G(\mu_t^h, \mu_t^o, y_0)$, we can perturb the generated human and object motion at

---

[2]We note that APDM and DBDM work independently. We thus use two symbols to denote the different diffusion time steps to avoid confusion.

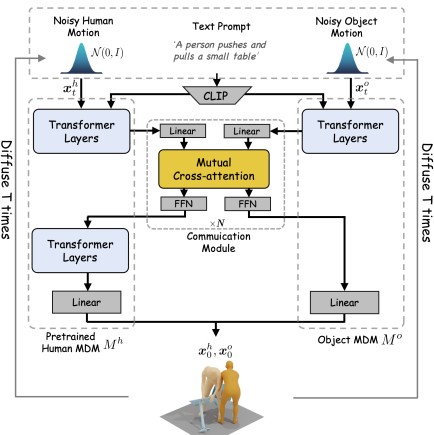

Figure 3. **Illustration of DBDM architecture for coarse 3D HOIs generation.** It has two branches designed for generating human and object motions individually. A mutual cross-attention is introduced to allow information exchange between two branches to generate coherent motions. The human motion model $M^h$ finetunes a pretrained MDM [51].

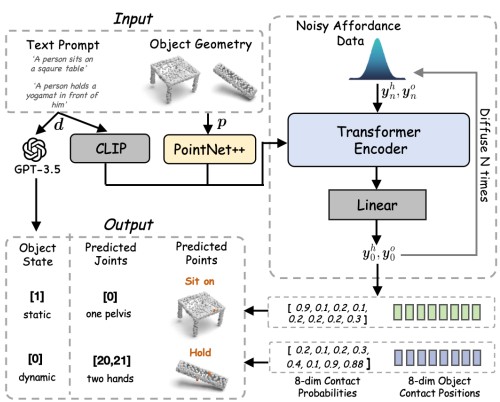

Figure 4. **Illustration of APDM architecture for affordance estimation.** Affordance information of human contact labels, object contact positions, and binary object states are represented together as a noise variable, which is fed into the Transformer encoder to generate clean estimation. The object point cloud and textual prompt are taken as conditional input.

each diffusion step $t$ as in [21, 60],

$$\boldsymbol{\mu}_t^h = \boldsymbol{\mu}_t^h - \tau_1 \Sigma_t \nabla_{\boldsymbol{\mu}_t^h} G(\boldsymbol{\mu}_t^h, \boldsymbol{\mu}_t^o, \boldsymbol{y}_0), \quad (3)$$

$$\boldsymbol{\mu}_t^o = \boldsymbol{\mu}_t^o - \tau_2 \Sigma_t \nabla_{\boldsymbol{\mu}_t^o} G(\boldsymbol{\mu}_t^h, \boldsymbol{\mu}_t^o, \boldsymbol{y}_0). \quad (4)$$

Here $\tau_1$ and $\tau_2$ are different strengths to control the guidance for human and object motion, respectively. Due to the sparseness of object motion features, we assign a larger value to $\tau_2$ compared to $\tau_1$. This applies greater strength to perturb object motion, facilitating feasible corrections for contacting joints. During the denoising stage, to eliminate diffusion models' bias that can suppress the guidance signal, we iteratively perturb $K$ times in the last denoising step. The details are illustrated in Algorithm 1 of Appendix.

How can we define the objective function $G(\boldsymbol{\mu}_t^h, \boldsymbol{\mu}_t^o, \boldsymbol{y}_0)$? We consider three terms here. First, in the generated 3D HOIs, the human and object should be close to each other on the contacting points. We therefore minimize the distance between human contact joints and object contact points

$$G_{con} = \sum_{i \in \{1,2,\ldots,8\}} \left\| R(\boldsymbol{\mu}_t^h(i)) - V(\boldsymbol{\mu}_t^o, \boldsymbol{y}_t^o(i)) \right\|^2, \quad (5)$$

where $\boldsymbol{\mu}_t^h(i)$ and $\boldsymbol{y}_t^o(i)$ denote the $i$-th available contacting joint indexed by $\boldsymbol{y}_0^h$ and $i$-th object contact point, respectively. $R(\cdot)$ converts the human joint's local positions to global absolute locations, and $V(\cdot)$ obtains the object's contact point sequence from the predicted mean of object pose $\boldsymbol{\mu}_t^o$.

Second, the generated motion of dynamic objects typically follows human movement. However, we observe that when the human interacts with a static object, such as sitting on a chair, the object appears slightly moved. To address this, we immobilize the object's movement in the generated samples if the state is static ($\boldsymbol{y}_0^s = 1$), ensuring that proper contact is established between the human and the static object. The objective is defined as

$$G_{sta} = \boldsymbol{y}_0^s \cdot \sum_{l=1}^{L} \left\| \boldsymbol{\mu}_t^o(l) - \bar{\boldsymbol{\mu}}_t^o \right\|^2, \quad (6)$$

where $\boldsymbol{\mu}_t^o(l)$ denotes the object's 6DoF pose in the $l$-th frame. $\bar{\boldsymbol{\mu}}_t^o = \frac{1}{L} \sum_l \boldsymbol{\mu}_t^o(l)$, which is the average of predicted means of the object's pose.

Third, we define a smoothness term $G_{smo}(\mu)$ for the object motion to mitigate motion jittering during contact. Due to the space limit, we explain it in Appendix A.3.

Finally, we combine all these goal functions to as the final objective

$$G = G_{con} + \alpha G_{sta} + \beta G_{smo}, \quad (7)$$

where $\alpha = 500$ and $\beta = 100$ are weights for balance.

# 4. Experiments

## 4.1. Setup

**Dataset.** Since the data designed for studying text-driven 3D HOIs generation is severely scarce, we manually label interaction types, interacting subjects, and contact body parts on top of the BEHAVE dataset [4]. We then use GPT-3.5 [34] to rephrase and generate three text descriptions for each HOI sequence, increasing the diversity of the data. Specifically, BEHAVE encompasses the interactions of 8 subjects with 20 different objects. It provides the human SMPL-H representation [29], the object mesh, as well as its 6DoF pose

information in each HOI sequence. To ensure consistency in our approach, we follow the processing method used in HumanML3D [13] to extract representations for 22 body joints. All the models are trained to generate $L = 196$ frames in our experiments. In the end, we have 1451 3D HOI sequences along with textual descriptions to train and evaluate our proposed approach. We follow the official train/test split on BEHAVE. We provide more details of the dataset and annotation process in Appendix I.

In addition, we evaluate our approach on OMOMO dataset [27]. OMOMO focuses on full-body manipulation with hands. It consists of human-object interaction motion for 15 objects in daily life, with a total duration of approximately 10 hours. It provides text descriptions for each interaction motion. We utilize their object split strategy for both training and evaluation, ensuring the objects between the training and testing sets are different. Additionally, we preprocess human and object motion, similar to our way for the BEHAVE dataset. More details are in Appendix J.

**Evaluation metrics.** We first assess different models for human motion generation using standard metrics as introduced by [13], namely *Fréchet Inception Distance (FID)*, *R-Precision*, and *Diversity*. *FID* quantifies the discrepancy between the distributions of actual and generated motions via a pretrained motion encoder. *R-Precision* gauges the relevance between generated motions and their corresponding text prompts. *Diversity* evaluates the range of variation in the generated motions. Additionally, we compute the *Foot Skating Ratio* to measure the proportion of frames exhibiting foot skid over a threshold (2.5 cm) during ground contact (foot height < 5 cm).

To evaluate the effectiveness of HOIs generation, we report the *Contact Distance* metric, which quantitatively measures the proximity between the ground-truth human contact joints and the object contact points. Ideally, we should develop similar metrics, *e.g*, *FID*, to evaluate the *stochastic* HOI generation. However, due to the limited data available in BEHAVE [4], training a motion encoder would produce biased evaluation results. To mitigate this issue, we resort to user studies to quantify the effectiveness of different models. Details will be introduced later.

## 4.2. Comparisons with Existing Methods

**Baselines.** Our work introduces a novel 3D HOIs generation task not addressed by existing text-to-motion methods, which focus exclusively on human motion generation without accounting for human-object interactions. To compare with existing works, we mainly focus on evaluating human motion generation. We then design different variants of our models for comparing 3D HOIs generation. Specifically, we adopt the prominent text-to-motion methods MDM [51] and PriorMDM* [45] with the following settings. (a) MDM†: In this setup, we finetune the original MDM model [51] on the BEHAVE dataset [4] without object motion. (b) MDM*:

This variant involves adapting the input and output layers' dimensions of the MDM model [51] to accommodate the input of 3D HOI sequences. This adjustment allows for the simultaneous learning of both human and object motions within a singular, integrated model. (c) PriorMDM* [45]: We adapt the ComMDM architecture proposed in [45], originally designed for two-person motion generation, to suit our needs for HOIs synthesis by modifying one of its two branches for object motion generation. (d) InterDiff [61]: While Inter-Diff is not designed for text-driven synthesis of 3D HOI, we added text conditioning to InterDiff as the baseline. More details are in Appendix C.

**Quantitative Results.** Table 1-left reports the quantitative results on BEHAVE dataset [4]. Compared with the baseline methods, our full method achieves the best performance. Specifically, it achieves state-of-the-art results in both *FID*, *R-precision*, and *Diversity*, underscoring its ability to generate high-quality human motions in the context of coherently interacting with objects. The best *Contact Distance* also suggests that our approach can generate physically plausible HOIs, capturing the intricate interplay interactions between humans and objects. Table 1-right presents the quantitative results on the OMOMO dataset. We used the train/test split of the OMOMO dataset to evaluate the model's inference capacity on unseen objects, including the small table, white chair, suitcase, and tripod. Our method consistently outperforms other baselines by a considerable margin across all metrics. Notably, due to the distinctiveness of objects in the training and testing sets, the results indicate the effectiveness of our approach in *generalizing to unseen objects*, proving superior performance compared to other models. We also provide user study results, please refer to Appendix G for details.

**Qualitative Results.** We showcase qualitative comparisons, rendered with SMPL [29] shapes, between our approach and the baseline methods in Figure 5. It is observed that the generated HOI motion by other baselines lacks smoothness and realism, where the object may float in the air (*e.g*, the toolbox in Figure 5 (b)). Furthermore, these baseline methods struggle to accurately capture the spatial relationships between humans and objects (*e.g*, the chair in Figure 5 (e)). In stark contrast, our approach excels in creating visually appealing and realistic HOIs. Notably, it adeptly reflects the intricate details outlined in text descriptions, capturing both the nature of the interactive actions and the specific body parts involved (*e.g*, raising the trash bin with the right hand in Figure 5 (a)). For the same object, our method can generate diverse HOIs using different body parts and contact points, as shown in Figure 14 in Appendix.

## 4.3. Ablation Studies

We conduct extensive ablation studies in Table 2 and Figure 10 in Appendix to validate the effectiveness of different components. We summarize key findings below.

| Method | BEHAVE | | | | | | OMOMO | | | | | |
|---|---|---|---|---|---|---|---|---|---|---|---|---|
| | FID ↓ | R-precision (Top-3) ↑ | Diversity → | Contact Distance ↓ | Pene ↓ | Foot Skate Ratio ↓ | FID ↓ | R-precision (Top-3) ↑ | Diversity → | Contact Distance ↓ | Pene ↓ | Foot Skate Ratio ↓ |
| Real | 0.04 | 0.86 | 12.48 | - | - | - | 0.57 | 0.63 | 9.98 | - | - | - |
| MDM† | 6.77 | 0.34 | 10.81 | - | - | - | 12.28 | 0.23 | 5.56 | - | - | - |
| MDM* | 4.25 | 0.38 | 11.23 | 0.448 | 0.52 | 0.190 | 10.37 | 0.21 | 6.04 | 0.768 | 0.41 | 0.191 |
| PriorMDM* | 4.54 | 0.30 | 10.03 | 0.416 | 0.57 | 0.270 | 9.87 | 0.25 | 6.34 | 0.523 | 0.38 | 0.344 |
| InterDiff | 8.58 | 0.26 | 10.75 | 0.506 | 0.42 | 0.218 | 14.27 | 0.17 | 5.69 | 0.906 | 0.32 | 0.239 |
| **Ours** | **1.62** | **0.46** | **12.02** | **0.347** | 0.51 | **0.182** | **8.76** | **0.31** | **8.13** | **0.326** | 0.39 | **0.141** |

Table 1. **Quantitative results on the BEHAVE and OMOMO dataset.** We compare our method with baselines adapted from existing models. MDM†: fine-tune the original MDM [51] on the BEHAVE dataset without object motion. MDM*: adapting the input and output layers' dimensions of the MDM to accommodate both human and object motions. PriorMDM*: We adapt the ComMDM architecture proposed in Shafir et al. [45]. InterDiff: We add a CLIP encoder in Xu et al. [61] to support our task. The right arrow → means closer to real data is better. Chois [26]: We remove object waypoints to make a fair comparison.

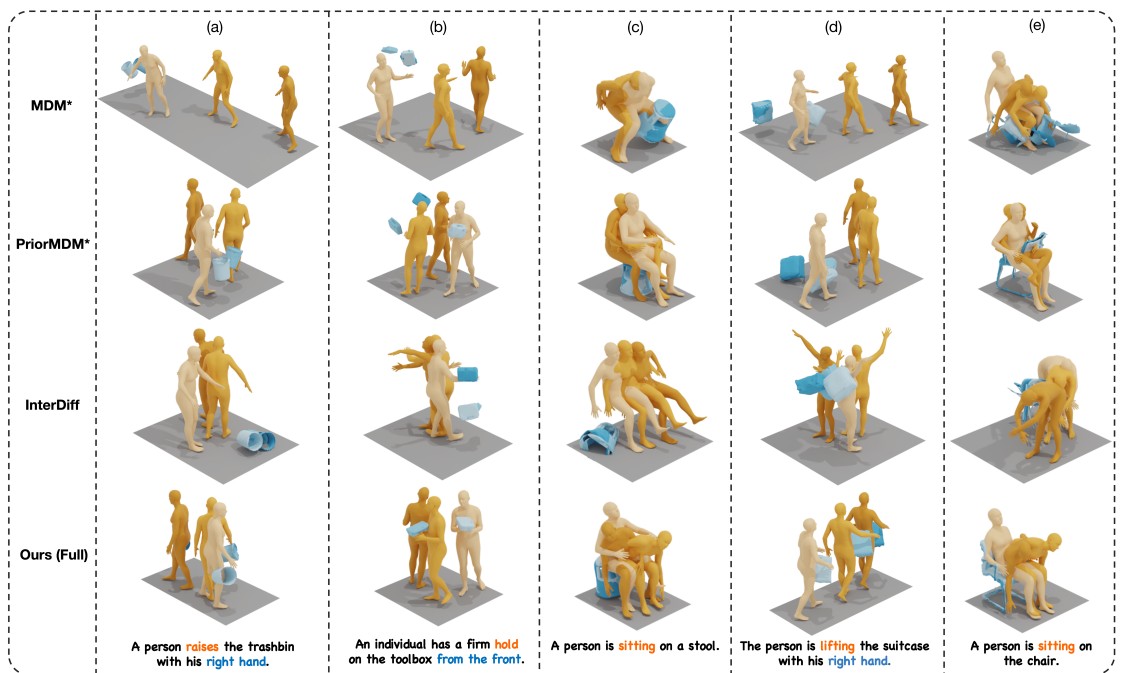

Figure 5. **Qualitative comparisons of our approach and baselines on BEHAVE dataset.** The bottom row, showcasing our method, demonstrates the generation of realistic 3D HOIs with plausible contacts, particularly evident in columns 2 and 4. This contrasts with the baselines, which fail to achieve a similar level of realism and contact plausibility in the interactions. As an additional visual aid, the mesh color gradually darkens over time to represent progression. (Best viewed in color.)

**Object MDM is helpful.** In Table 2, we compare *Ours w/o $M^o$ & CM* and *ours (Full)* to demonstrate the importance of the Object MDM. In *Ours w/o $M^o$ & CM*, we exclusively finetune the human MDM, while randomly initializing the object motion. The Communication Module (CM) is also ignored due to the removed object MDM. Interaction correction is then applied to optimize contact between the human and object. The interaction correction with random initial object motion produces worse results, demonstrating the importance of initial object motion from Object MDM.

**DBDM with Communication Module ($CM$) is critical.** In Table 2, we compare *Ours w/o CM* and *ours* to demonstrate the effectiveness of the Communication Module. When eliminating $CM$, the results drop substantially across all metrics, with a particularly significant decrease in *Contact Distance*. The visual results (w/o $CM$) in Figure 10 of Appendix further validate this point.

**Leveraging the pre-trained Human motion prior can generate better human motions.** We aim to utilize the strong motion prior from the pre-trained human motion model to enhance the realism of the generated motion. Table 2 (*Ours w/o pretrain*) reports the results of training human MDM from scratch, without resuming the weights from the pre-trained MDM [51]. Comparing *Ours w/o pretrain* and *Ours* demonstrates the effectiveness of leveraging the pre-trained MDM.

**Interaction Correction makes better HOIs generation.** In Table 2, we compare our full method (*Ours (full)*) to a vari-

| | BEHAVE | | | | | OMOMO | | | | |
|---|---|---|---|---|---|---|---|---|---|---|
| Variants | FID $\downarrow$ | R-precision (Top-3) $\uparrow$ | Diversity $\rightarrow$ | Contact Distance $\downarrow$ | Foot Skate Ratio $\downarrow$ | FID $\downarrow$ | R-precision (Top-3) $\uparrow$ | Diversity $\rightarrow$ | Contact Distance $\downarrow$ | Foot Skate Ratio $\downarrow$ |
| Real | 0.04 | 0.86 | 12.48 | - | - | 0.57 | 0.63 | 9.98 | - | - |
| *w/o Interaction Correction* | | | | | | | | | | |
| Ours w/o CM | 3.11 | 0.36 | 10.54 | 0.524 | 0.265 | 11.57 | 0.27 | 7.92 | 0.588 | 0.231 |
| Ours w/o pretrain | 2.98 | 0.39 | 11.21 | 0.402 | **0.158** | 10.38 | 0.29 | 7.82 | 0.412 | 0.167 |
| Ours$^{global}$ | 15.37 | 0.28 | 10.85 | 0.375 | 0.274 | 20.22 | 0.21 | 8.02 | 0.366 | 0.348 |
| Ours | 2.10 | 0.38 | 11.26 | 0.415 | 0.205 | 9.12 | 0.29 | 7.97 | 0.397 | 0.193 |
| *w/ Interaction Correction* | | | | | | | | | | |
| Ours w/o $M^o$ & CM | 3.93 | 0.32 | 11.43 | 0.365 | 0.310 | 11.03 | 0.28 | 7.98 | 0.536 | 0.331 |
| Ours $^{joint}$ | 4.37 | 0.31 | 11.25 | 0.421 | 0.342 | 11.52 | 0.27 | 7.92 | 0.547 | 0.325 |
| Ours w/o $G_{con}$ | 2.02 | 0.37 | 11.97 | 0.417 | 0.196 | 9.23 | 0.28 | 8.03 | 0.332 | 0.144 |
| Ours w/o $G_{sta}$ | 1.81 | 0.39 | 11.54 | 0.367 | 0.181 | 9.11 | 0.30 | 8.10 | 0.340 | 0.142 |
| Ours w/o $G_{smo}$ | 1.83 | 0.41 | 11.67 | 0.370 | 0.182 | 8.98 | 0.29 | 8.06 | 0.345 | 0.142 |
| Ours (Full) | **1.62** | **0.46** | **12.02** | **0.347** | 0.182 | **8.76** | **0.31** | **8.14** | **0.326** | **0.141** |

Table 2. **Ablation studies of our model's variants on the BEHAVE and OMOMO datasets.** The right arrow $\rightarrow$ means closer to real data is better. *w/o CM*: we remove the Communication Module (CM) in the DBDM model. *w/o pretrain*: we train human MDM from scratch on BEAHVE dataset. *global*: we adopt the global human pose representation proposed by Liang et al. [28] for both the pretraining of human MDM and the finetuning of DBDM. *w/o $M^o$ & CM*: We exclusively finetune the human MDM, while randomly initializing the object motion. Interaction correction is then applied to optimize contact between the human and object. *joint*: We train a single diffusion model that jointly generate human motion, object motion, and affordance. *w/o $G_{con}/G_{sta}/G_{smo}$*: without contacting/static/smoothness goal function in interaction correction.

ant without interaction correction (*Ours*) to demonstrate the effectiveness of interaction correction. The model with interaction correction consistently outperforms the variant across all control accuracy metrics. As shown qualitatively in Figure 10 of Appendix, our full method produces more realistic HOIs with better contact compared to the model without interaction correction. Furthermore, all sub-functions in Interaction Correction contribute to the realistic HOI generation, as demonstrated in *Ours w/o $G_{con}$, w/o $G_{sta}$, w/o $G_{smo}$* of Table 2.

**Why Human MDM and Object MDM are needed separately?** We can ablate this by comparing Table 1 (*MDM\**) and Table 2 (*Ours (w/o Interaction Correction)*. In *MDM\** we jointly learn both human and object motion with a diffusion model. Our superior results demonstrate that separately modeling human motion and object motion with a communication module can achieve better results. A key advantage is that the human motion diffusion model (MDM) can finetune a pre-trained MDM [51], leveraging the extensive prior knowledge from the large-scale HumanML3D dataset. In contrast, jointly predicting human and object motion with a single transformer requires training from scratch (due to the change of the model architecture) on the much smaller BEHAVE dataset, which results in poorer human motion results.

| | AP (%) $\uparrow$ | L2 Dist $\downarrow$ |
|---|---|---|
| Ours $^{joint}$ | 53.67 | 0.384 |
| Ours $^{APDM}$ | **78.54** | **0.272** |

Table 3. APDM evaluation. The reported metrics include Average Precision (AP) for predicted human contact probabilities and L2 Distance (Dist) error for predicted object contact points.

**Why not jointly generate motion and affordance with one unified model?** We attempt to generate human motion, object motion, and affordance jointly within the same model, as indicated in the Table 2 (*Ours$^{joint}$*). Our joint prediction concatenates affordance data with motion data along the channel dimension and adjusts the input and output dimensions of MDM to generate motions and affordance simultaneously. Comparing Table 2 *Ours$^{joint}$* and *Ours (full)* demonstrates that our modular design significantly improves human motion quality, as evidenced by metrics such as FID, R-Precision, and Foot Skate Ratio, as well as the interaction quality measured by Contact Distance. Table 3 further validates that our modular design achieves more accurate affordance estimation, measured by AP and L2 Distance. The improvement is attributed to the fact that affordance learning is highly dependent on the geometry of 3D data and text semantics, rather than human and object motions. Therefore, disentangling these elements enhances their respective performances.

## 5. Conclusion

In summary, we presented a novel approach HOI-Diff to generate realistic 3D HOIs driven by textual prompts. By employing a modular design, we effectively decompose the complex task of HOI synthesis into simpler sub-tasks, enhancing the coherence and realism of the generated motions. Our HOI-Diff model successfully generates coarse dynamic human and object motions, while the affordance prediction diffusion model adds precision in predicting contact areas. The integration of estimated affordance data into classifier-guidance further ensures accurate human-object interactions. The promising experimental results on our annotated BEHAVE dataset demonstrate the efficacy of our approach in producing diverse and realistic HOIs.

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
