# OpenReview forum: "HOI-Diff: Text-Driven Synthesis of 3D Human-Object Interactions using Diffusion Models"
_thecvf.com/CVPR/2025/Workshop/HuMoGen — CVPR 2025 Workshop HuMoGen Submission_

### Official Review · Reviewer_ih1f · 2025-03-26
**impressive results with extensive evaluation**

**Rating:** 5
**Confidence:** 5

**Review:**

This paper presents a new method for generating human-object interactions based on text input. While the individual components are based on existing work, such as human motion synthesis using MDM, combining them to generate high-quality results is a great effort and the authors also proposed the contact point prediction and refinement in this work. Since human-object interactions can be applied to a wide range of applications such as interactive graphics and games, this work will benefit a significant number of researchers in the community. Furthermore, the authors will share the annotated data to further extend the impact of the work.

The paper is well-written in general, and there is also detailed supplementary material attached. Extensive quantitative and qualitative results are presented to show the pros and cons of the work.

There is a minor point for the authors - under Section 3.3, it is not very clear how those 8 contact points on the object were presented. For example, are those trained using the contact points from the GT data? Does it always have 8 contact points? More explanation under Section 3.3 will help.

---

### Official Review · Reviewer_hGRX · 2025-03-27
**Novel and impactful design decisions with great results for human-object interactions generation**

**Rating:** 5
**Confidence:** 5

**Review:**

This paper presents a novel dual-branch diffusion model to generate both human and object motions conditioned on the given text description for the task of HOI generation. It also introduces an affordance prediction model to estimate the contact region between humans and objects, which could be further used to refine the HOI generation results. Empirical experiments demonstrate the effectivenss of the proposed method against SOTA methods.

Strengths:
1. The proposed dual-branch diffusion model to model the human and object motion in separated branches is novel and makes sense to me. Extensive ablation studies also well support the effectiveness of this design. Moreover, it enables HOI model to make use of pretrained single-person MDM which is helpful.
2. The video results look very good.
3. I appreciate that the author promised to open-source their data annotation and code, which would be definitely useful for the community.
4. If I'm not wrong, the proposed affordance prediction model is very helpful and has already been widely adopted in later works to effectively improve interaction quality as a post-processing step.

Weaknesses:
There's no major concern for this paper. Trying to include more comparison to more concurrent works would be appreciated, but anyway, this proposed method in this paper is well-designed and well-presented.

---

### Meta-Review · Area_Chair_FK2g · 2025-03-29

**Recommendation:** Accept

**Metareview:**

Both reviewers unanimously rate this as a strong accept (5/5) with the highest confidence level (5/5), indicating exceptional quality and significance. The reviewers, who are familiar with the relevant literature, recognize the paper as an important contribution to HOI generation.
While the individual components build upon existing work (e.g., human motion synthesis using MDM), the integration of these components into a cohesive framework for HOI generation represents a valuable contribution. The affordance prediction model is highlighted as particularly impactful, having already influenced subsequent research.
The consensus is that this paper introduces a technically sound, well-evaluated approach to an important problem, with clear practical applications in interactive graphics, games, and related domains. The commitment to open-source data and code further enhances its value to the research community.

---

### Decision · Program_Chairs · 2025-03-31

Accept